# Do Hens Use Enrichments Provided in Free-Range Systems?

**DOI:** 10.3390/ani12080995

**Published:** 2022-04-12

**Authors:** Victoria Sandilands, Laurence Baker, Jo Donbavand, Sarah Brocklehurst

**Affiliations:** 1Scotland’s Rural College (SRUC), Easter Bush Campus, Midlothian EH25 9RG, UK; laurence.baker@sruc.ac.uk (L.B.); jo.donbavand@sruc.ac.uk (J.D.); 2BioSS, JCMB, The Kings Buildings, Peter Guthrie Tait Road, Edinburgh EH9 3FD, UK; sarah.brocklehurst@bioss.ac.uk

**Keywords:** hay, pecking blocks, scattered feed, ropes, injurious pecking, feather scores, cost, behaviour, welfare

## Abstract

**Simple Summary:**

Free-range hens are typically given enrichments to encourage foraging and reduce injurious pecking. Of four enrichments (Lucerne hay, pecking blocks, pelleted feed scattered in litter, and jute ropes) provided to eight commercial flocks of free-range hens, pecking blocks and bales provided consistent interest to hens, based both on observations of hens in the vicinity of the enrichments (doing anything), interacting with the enrichments, and least walking/running or standing near the enrichments. Hens were most interested in pelleted feed at the time of scatter, but pelleted feed was consistently of greater interest than ropes, which hens seemed to find least attractive. Ropes were no more attractive to hens than no enrichment at all. Feather scores (a proxy measure for feather pecking) worsened with age, but differences between treatments were small and variable between ages, possibly due to lack of data and/or hens mixing between treatments. While ropes were by far the cheapest enrichment to provide, behaviour at ropes was indistinguishable from behaviour away from any enrichments, and thus did not sufficiently encourage foraging and other desirable behaviours. A balance between encouraging positive hen behaviour and cost to the producer needs to be taken into account in the practical use of any enrichment.

**Abstract:**

Hens in free-range systems are given enrichments to increase foraging and limit injurious pecking, but the efficacy of enrichment types requires investigation. We studied hen behaviour and feather cover in eight commercial free-range flocks each given access to four enrichments within the shed. Sheds were split into quarters, in which two enrichments (jute ropes (R) + other) were installed. Other enrichments were: lucerne hay bales (B), pecking blocks (PB), pelleted feed (PF), or further R (control). Hens were observed at three ages, at three times per age (−1, 0, ≥1 h relative to PF application), in 1 m diameter circle locations around ropes (ControlR), Enrich (B, PB, PF, R), and Away from each enrichment. Feather scores were recorded at all ages/times, at the Away location only. Significantly more birds were in Enrich locations where PB, B, and PF were available, and least near R, ControlR, and Away locations (*p* < 0.001). Proportions of birds interacting with enrichments were significantly higher for PB, B, and PF than R (*p* < 0.001), but enrichments did not generally affect proportions of birds foraging in the litter, apart from a significant decrease (*p* < 0.001) in PF birds foraging in the Enrich location because they were directing behaviour at PF instead. Feather scores worsened with age (*p* < 0.001) but were not consistently affected by enrichment. Enrichment replacement rates varied between farms. Enrichments costs were highest for PB and cheapest for R. Enrichments except R were used by hens, but with no obvious effect on feather cover. A balance has to be struck between enrichment benefits to hens and economics, but evidence suggested that hens did not benefit from R.

## 1. Introduction

The use of enrichments in captive animal housing is commonplace, with the main aim to improve animal welfare. Newberry [1] defined environmental enrichment as modifications to the environment that improve the biological functioning of captive animals, such as by improving health, which is one of the key components of good welfare. Injurious pecking, which includes gentle and severe feather pecking, cannibalism, and vent pecking [2] is one such laying-hen behaviour that damages the health of its victims. Feather pecking, the most common form of injurious pecking [3], is thought to be redirected foraging (i.e., food seeking) behaviour [4,5,6]. In alternative (e.g., barn and free-range egg production) systems in the UK and EU, one-third of the floor area must be litter (which is defined as any friable material that enables hens to show natural behaviour) [7]; commonly, this is provided as sawdust, wood shavings, straw, or a mixture [3]. While litter provides a foraging substrate, and is undoubtedly better than no substrate at all (for a review, see [8]), there is likely to be little positive feedback from litter that consists of only non-nutritive material and bird faeces. In addition, pressure to ban beak trimming as a means of controlling damage when feather pecking develops means that there is even greater interest in preventing the behaviour from starting. Therefore, increasing appropriate pecking behaviour by providing pullets and laying hens with suitable enrichments to peck at inside of loose-housed sheds is becoming more commonplace in these systems. While providing enrichments in these systems does not guarantee reduction of feather pecking and plumage damage, in many instances they have a positive effect (for a review, see [9]).

It is generally accepted that indestructible items such as balls, cones, and hanging CDs are unsuitable for encourage foraging behaviour in laying hens. They lack fundamental characteristics that elicit foraging; namely, that they do not fall apart on pecking, and there is no nutritional benefit to them [1,6]. That said, commercial free-range systems do use hanging ropes as enrichment [10], which are manipulable and fray, although they are non-nutritive, and have been demonstrated to reduce feather pecking in cages and floor pens in some studies [11] and to have no effect in others [12]. Previous work has shown that substrates that are destructible and provide nutrients are more likely to improve foraging and/or reduce injurious pecking than those that are/do not [6,9,13,14]; however, this is not consistently true (e.g., [15,16]). Destructible enrichments include long-cut straw, pecking blocks/stones, alfalfa hay bales, silage, and carrots [14,16,17,18]. Nutritive enrichments can potentially have an impact on feed consumption, egg production, and/or egg quality. While Schreiter et al. [14] found no effect of alfalfa hay or pecking stones on daily feed consumption, body mass, or egg-shell-breaking strength, they did find that these affected albumen quality. Steenfeldt et al. [17] found egg production increased with two out of three foraging materials provided, but intake of these was extremely high (33–48% of total feed intake). In contrast, Cronin et al. [19] found no effect of straw enrichment on laying performance.

The aim of this study was to investigate hen use of four commonly provided destructible enrichments in commercial free-range flocks, and their effects on feather scores as a proxy measure of feather pecking. We hypothesised that hens provided with rope-only enrichment would have the worst feather cover and show the least foraging behaviour around the enrichment. We also estimated the cost of enrichments, based on replacement rates seen.

## 2. Materials and Methods

This work was approved by SRUC’s animal ethics committee (study number AU AE 36-2018, approved on 30 May 2019).

### 2.1. Housing and Birds

Eight flocks (A–H) of free-range hens at four different farms (2 flocks/farm) were recruited for the study via British Free Range Egg Producers Association (BFREPA) membership. All farms were based in Scotland, and pullets arrived at the farms at 15–16 weeks of age. The free-range sheds contained multitier structures (all Natura Step, Big Dutchman International GmbH, Vechta, Germany) of three levels (the litter floor and two tiers), and were split into four 4000-bird colonies (‘quarters’) by high fencing installed within the shed. Quarters were identified as Q1 (nearest the annex where staff first entered) to Q4 (Figure 1). Hens could access the range area through popholes in the outer wall of the shed, but the range area was not split into quarters, thus hens could potentially exit one quarter and enter another. There were 16,000 hens in total per flock. Normal commercial practice was undertaken at each farm, apart from the provision of enrichments. The bird strains used were all brown egg layers (Lohmann Brown, 3 flocks; Bovan Brown, 3 flocks; H&N Brown Nick, 2 flocks).

### 2.2. Enrichment Treatments

Some schemes for egg production assurance require that at least two enrichments are provided for every 1000 hens, some of which must be destructible (e.g., RSPCA, 2017). Therefore, we used this level and these types of enrichments in our study, all provided inside the sheds. All flocks were provided with 2 different enrichment items per 1000 hens, and thus 8 enrichments per quarter. Enrichments were installed shortly after pullets arrived at the laying farm, and farm staff were advised of a replacement schedule based on the estimated time each enrichment should last; however, staff were advised to replace as often as necessary to ensure that hens were never without the enrichments (apart from pelleted feed, which was always given twice a day—see below). There were four different enrichments used (i–iv below, and Figure 2), all of which were destructible:Lucerne (alfalfa) hay bales. Analysed content: 16.7% crude protein, 90.4% dry matter. Four bales provided per quarter (1 per 1000 hens), which were placed into hay nets and suspended over the litter (some farms placed them on the floor initially, and hung them up after approximately 3 days). Bales weighed approximately 15 kg and measured 65 × 45 × 35 cm or 102,375 cm^3^ per bale. Cost: from GBP 6.50 per bale. Estimated to last 3 weeks.Pecking blocks (Pickblock^TM^ medium, Crystalyx^®^ Products GmbH, Münster, Germany), compact hard edible blocks made of grains (rye, maize, wheat), calcium carbonate, oyster shells, dextrose, molasses, wheat gluten feed, and lucerne meal; crude protein 5.8%; weight 5 kg; dimensions 23 × 16.5 × 13 cm, or 4934 cm^3^ per block. Provided at 1 block per 500 hens, thus 8 blocks per quarter, which were placed in pairs onto slats or plastic bucket lids (to stop them from getting damp) on top of the litter. Cost: approximately GBP 7 per block. Estimated to deteriorate at 1 g/hen/day, and thus expected to last approximately 10 days.Pelleted feed formulated for layers (Farmgate Layers Pellets, ForFarmers UK Ltd., Dumfriesshire, UK). Analysed content: 16.0% protein, 86.2% dry matter. Provided 2 kg twice a day, scattered from side-to-side covering a roughly 0.5 m width, down the centre of the litter area (Figure 1), thus providing 1 g pellets/hen/day. Staff were provided with plastic jugs marked with a ‘fill’ line to the correct weight, and feed was stored in plastic bins within the shed quarter for ease of use and rodent control. The timing of scattering was arranged to coincide with staff inspections/collections of floor eggs, and ranged from farm-to-farm between 09:00–11:30 (scatter 1), and 13:00–16:30 (scatter 2). Cost: GBP 8.38 per 20 kg bag, or GBP 419/tonne. Estimated to last up to a few hours.Jute ropes (Ropes Direct, Norfolk, UK). Four ropes (8 mm diameter, cut into 30 cm lengths and looped in half; approximately 15.1 cm^3^ in volume per rope) were attached initially by polypropylene string (flocks A, B), and then cable ties (all flocks) to the first platform or alighting rails of the multitier structure, evenly spread along the structure. Cost: just over GBP 0.08 per 30 cm, or GBP 0.33 for 4 rope pieces. Estimated to last 6 months.

The rope was considered the standard (control) enrichment, so combinations of enrichment treatments (known as classification factor ‘**treatment**’) for each quarter were:4 bales and 4 ropes (B);4 pairs of pecking blocks and 4 ropes (PB);4 kg pelleted feed and 4 ropes (PF);8 ropes (R).

Due to their predicted destruction/intake rate, edible enrichments (e.g., B, PB, and PF) were not expected to have an influence on feed intake or egg production and egg quality (none of which was measured here). Enrichments were offered in a balanced design over all quarters with all treatments provided to each flock, by placing enrichments in each shed based on two Latin squares (Table 1).

### 2.3. Behaviour Observations and Feather Scores

Observations were due to take place during three visits at 34, 52, and 70 weeks of age (i.e., every 18 weeks) (known as classification factor ‘**age**’). Actual flock visits took place when birds were 33 weeks 5 days–34 weeks 6 days old, 51 weeks 6 days–53 weeks old, and 70 weeks 1 day–71 weeks 4 days old, but for simplicity, they are still referred to as visits at 34, 52, and 70 weeks of age throughout. The two flocks on a single farm were observed on two consecutive days, by one of two people. Popholes were open during observations. Bird behaviour was recorded using scan-sampling methods, at times relative to the first scatter of pelleted feed: −1, 0, and 1 h (known as classification factor ‘**time**’). The observer always began in the quarter with the pelleted feed treatment and then moved up the quarters (e.g., if PF was in Q2, then they observed in the order Q2, Q3, Q4, Q1). The observer entered the shed quarter and positioned themselves between the treatment enrichments (B, PB, PF, R) and the control enrichment (R), and remained quiet for 3 min to allow the hens to settle. The observer then scan-sampled a 1 m diameter area around three locations: the treatment enrichment (‘Enrich’), the control (ropes) enrichment (‘ControlR’), plus a 1 m diameter area away from either enrichment (‘Away’) (known as factor ‘**location**’). For R, both the ‘Enrich’ and the ‘ControlR’ observations were at ropes. A count of birds within each of the three circles and their behaviours (Table 2) were recorded.

The counts were repeated three times in 15 min (e.g., at 3, 8, and 13 min). Thus, a total of 324 observations per flock were made (i.e., 3 locations × 3 observations per time relative to scatter × 3 times relative to scatter × 4 quarters × 3 ages). The observer moved to the next quarter after 15 min, so that all four quarters per flock were observed within each 1 h period.

Feather scores (i.e., the recording of feather damage on a scale of 0–5, where 0 = no damage, 1 = slight damage/loss with no bare skin, up to 5 = 1–2 cm^2^ haemorrhage or >5 × 5 cm^2^ bare skin with <1 cm^2^ haemorrhage [20]) of five body locations (neck, back, tail, breast, and both wings) were carried out remotely (i.e., without handling, [21]) on 10 birds in the Away location once per scan sampling time (−1, 0, 1) per treatment (i.e., quarter) at each age (thus 5 feather scores/bird × 10 birds/time × 3 sampling times × 4 treatments × 3 ages = 1800 scores/flock).

Due to a combination of heightened biosecurity related to avian influenza and COVID-19, some visits to flocks were prevented. As a result, no feather scores or behaviour data were collected at age 52 weeks for flocks G and H, and no behaviour data were collected at age 70 weeks for flocks C, D, E, and F. Feather scores for C, D, E, and F at age 70 weeks were recorded from photographs taken by the farm staff of the birds in the Away location, from 10 birds. However, data from photos were judged to be unreliable, as they did not follow patterns seen in other flocks, with higher scores than expected at some body locations and lower than expected at other body locations. Therefore, the data from photos were omitted from all means shown and analyses.

### 2.4. Statistical Analyses

Behaviour data were analysed with the following fixed effects: age (34, 52, 70 weeks), time (−1, 0, 1 h), location (ControlR, Enrich, Away), and treatment (R, B, PB, PF) (and their interactions). For R, both the ‘Enrich’ and the ‘ControlR’ observations were at ropes, so one was randomly assigned to ControlR and one to Enrich to give the full complement of three locations to allow a full factorial statistical analyses of behaviour data. Random effects were flock, shed quarter, location within shed quarter, and interactions of these spatial effects with age, time within age, and scans within time within age, but many of these effects were negligible, and so were dropped from some models in order to achieve convergence.

With hen behaviour, three elements were analysed:Total counts of birds in each location (ControlR, Enrich, Away) at a scan engaged in all behaviours (because total birds in a particular location might indicate a desire to be there);Counts of birds engaged in each particular behaviour in each location at a scan;Proportions of birds engaged in each behaviour (i.e., counts of birds performing a behaviour/total birds in that location per scan).

Results for counts of birds engaged in particular behaviours are not shown because results were similar for counts and proportions.

Feather scores were summed over all body sites per bird, and total feather score was analysed. Fixed effects were age (34, 52, 70 weeks), time relative to scatter (−1, 0, 1), treatment (R, B, PB, PF) (and their interactions). Random effects were flock, shed quarter, and interactions of these spatial effects with age and time within age. Analyses focused on total feather scores, but some analysis is also reported from analysing feather scores from individual sites using LMMs fitted to feather scores (not transformed) or GLMMs applied to a binary data feather score >0, adding site and interactions with site to the fixed effects.

To analyse proportions, generalised linear mixed models (GLMMs) were fitted to binomial counts with appropriate binomial totals, logit link function (i.e., for proportion *p*, log_e_ (*p*/1 − *p*)), binomially distributed errors, and dispersion fixed at 1. To analyse counts, GLMMs were fitted to the counts with log link function, Poisson distributed errors, and dispersion fixed at 1. Where data were sparse, GLMMs with all effects included would not converge, so random and fixed effects in these models were simplified. Linear mixed models (LMMs) with all effects included were fitted to the total feather score after log transformation (i.e., log_e_ (total feather score + 1)) and were used as approximations in addition to simplified GLMMs for binomial data and counts. With LMMs, proportion data were first angular-transformed to a degrees scale (see Equation (1) below) to normalise the distribution of residuals; i.e., for proportion *p*:(1)$$\frac{180}{\pi }sin^{-1}\left(\sqrt{p}\right)$$

While counts and total feather scores were natural log transformed. Where high-level interactions were substantial, lower-level effects are not reported.

Due to the large number of tests being carried out, the results focus on highly significant effects and make clear when results were marginal. In some instances where interactions were marginal, lower-level associated effects are also shown. The *p*-values were based on approximate *F* tests when available, but otherwise were based on Wald tests; statistics for *F* tests are given in the results as *F***_ndf,ddf_**, where ndf is the numerator degrees of freedom (the number of effects to be estimated, which is the number of levels for a categorical factor less 1) and ddf is the denominator degrees of freedom; or for Wald tests as Wald**_ndf_**/ndf to make this comparable with the *F* statistic. Tables and figures either show raw means along with standard deviations (SDs) or model estimates ± standard errors (SEs) obtained from the LMMs and GLMMs as well as these estimates back transformed onto the original scale (e.g., proportions or counts) to aid interpretation.

For replacement of enrichments, the mean and SD over flocks (*n* = 8) of the mean days between replacement of each enrichment per flock was calculated. We also briefly investigated the above-reported statistical models of behaviour data, including covariates on days since last replacement and the cumulative amounts of enrichments replaced at each visit, generating *p*-values for the covariates tested last after all other fixed effects and examined estimated coefficients. All data were compiled and linked in Excel. Genstat 18 was used for data processing and all statistical analyses.

## 3. Results

Mean mortality across flocks was 4.8% (range: 2.60–7.98%). Observation times relative to scatter feed application were in reality 1.5–0.47 h before scatter (still called −1 h for simplicity), 0.0–0.17 h (0 h) at scatter, and 1.0–3.0 h postscatter (hereafter referred to as ≥1 h).

Overall mean proportions of birds observed in behaviours, according to location and treatment, are shown in Table 3. In the area where only ropes were available (ControlR) and in the Away location, most hens were observed standing/sitting, followed by foraging and walking/running. Hens observed in ControlR showed low proportions of birds interacting with the enrichments (ropes). In the Enrich location, the mean proportions of hens in R treatments were mostly standing/sitting, whereas with other treatments, much higher proportions of birds were interacting with the enrichments. All proportions of hens observed in dustbathing, feather pecking, perching, and other were low.

### 3.1. Counts of Birds (Over All Behaviours)

On average over scans, there were 6.8–17.2 hens observed per location by treatment (Table 4).

There was a highly significant interaction between time, location, and treatment in the total numbers of birds observed over all behaviours (Wald_12_/ndf = 4.62 by GLMM, *p* < 0.001) (Figure 3a). There were more birds in the Enrich locations when the enrichments were not R, with the most birds observed with PB, then B, then PF. When feed was scattered (time 0), the number of birds went up only for PF in the Enrich location (and correspondingly went down for PF at the ControlR and Away locations, as hens moved away from these areas to the enrichment area), and then returned to −1 levels by time ≥1. In contrast, the numbers of birds in all locations with PB, B, and R remained constant across the three observation times.

The total numbers of birds observed, regardless of location, were similar between the different treatments at age 34 weeks, but treatment differences increased with age; at age 70 weeks, the greatest number of birds were observed for PB and the least for R (Figure 3b) (marginally significant interaction age.time.treatment, Wald_12_/ndf = 1.94 by GLMM; *p* = 0.026). Other effects of bird age were also marginal, but on average, the total birds observed declined with age at all locations (interaction of age.location, Wald_4_/ndf = 2.69 by GLMM; *p* = 0.030) (Table 5).

### 3.2. Behaviour

#### 3.2.1. Interacting with Enrichments (ControlR and Enrich Locations Only)

Of the total birds observed, the mean proportion of birds interacting with the enrichments in the Enrich locations was higher for PF at scatter-feeding time (0), then PB, then B, and was lowest for R (highly significant interaction time.location.treatment; *F*_6,621_ = 8.44 by GLMM; *p* < 0.001); however, proportions were consistent across all three times for PB and B, whereas interaction with PF dropped at times −1 and ≥1 (Figure 4). Observations of birds in all treatments in the ControlR locations, plus R birds in the Enrich location, showed similarly low proportions of birds interacting with R, compared to B, PB, and PF birds in the Enrich location. All interactions with bird age, and the main bird age effect, were not statistically significant for the mean proportion of birds (all *p* > 0.05).

#### 3.2.2. At (but Not Interacting with) Enrichments (ControlR and Enrich Locations Only)

The proportion of birds at, but not interacting with, the enrichments was highest with PF outside of scatter feeding time (i.e., at time −1 and ≥1), then PB, then B in the Enrich locations, with a much lower proportion for R (highly significant interaction time.location.treatment; *F*_6,122_ = 13.41 by LMM; *p* < 0.001) (Figure 5). The proportion of birds at, but not interacting with, the PB and B enrichments was consistent across all three observation times. There was a weak and inconsistent effect of age (marginally significant interaction location.age.treatment; *F*_6,63_ = 2.49 by LMM; *p* = 0.032) (data not shown). The other three-way interactions were not statistically significant.

#### 3.2.3. Stand/Sit

There were some marginally significant three-way interactions in stand/sit behaviour that were largely due to hens in PF treatment at location Enrich: the proportion of PF Enrich birds observed in stand/sit was both greatest at time ≥ 1 (time.location.treatment, Wald_12_/ndf = 1.92 by GLMM; *p* = 0.027) and lowest at age 34 weeks (age.location.treatment, Wald_12_/ndf = 2.15 by GLMM; *p* = 0.012) (data not shown). Averaged over other fixed effects, the proportion of birds observed in stand/sit behaviour increased with age (predicted means ± SE logit (back-transformed proportions) 34 weeks −0.80 ± 0.09 (0.31), 52 weeks −0.53 ± 0.10 (0.37), 70 weeks −0.28 ± 0.12 (0.43); Wald_2_/ndf = 8.69 by GLMM; *p* < 0.001). There was a highly significant interaction of location.treatment in the proportion of birds observed in stand/sit behaviour (Wald_6_/ndf = 34.94 by GLMM; *p* < 0.001): the greatest proportions of hens standing/sitting were seen in those locations where there were no enrichments (i.e., Away) or only rope enrichments (i.e., location ControlR, and treatment R in Enrich; whilst for PF, PB, and B, significantly fewer hens were standing/sitting at location Enrich (Figure 6).

#### 3.2.4. Forage

There was a weak three-way interaction of time.location.treatment on the proportion of birds observed foraging (excluding the PF scatter area) (Wald_12_/ndf = 1.81 by GLMM; *p* = 0.041) that was solely due to a decrease in PF birds foraging at litter (other than where feed was scattered) at time 0 in the Enrich location, but this was merely due to no birds foraging on anything other than PF scattered at this time (data not shown). There was a highly significant location.treatment interaction on the proportion of birds foraging (Wald_6_/ndf = 9.49 by GLMM; *p* < 0.001), again due to a decrease in PF birds foraging at litter (other than where feed was scattered) at time 0 (Figure 7). Foraging decreased with bird age (predicted means ± SE logit (back-transformed proportions) 34 weeks −1.64 ± 0.10 (0.16), 52 weeks −1.72 ± 0.10 (0.15), 70 weeks −2.09 ± 0.13 (0.11); Wald_2_/ndf = 8.42 by GLMM; *p* < 0.001).

#### 3.2.5. Walk/Run

The proportion of birds observed in walk/run behaviours was marginally affected by the interaction of time.location.treatment (Wald_12_/ndf = 2.10 by GLMM; *p* = 0.014) largely due to the influence of PF and time relative to scatter, for which walking/running declined then increased at the enrichment and commensurately increased then declined at ControlR; whilst for the other treatments, behaviour remained broadly steady with the times observed relative to scatter (Figure 8a). There was a highly significant location.treatment interaction on the proportion of birds observed in walk/run behaviour, where birds were observed walking/running least in the Enrich area with all treatments except R, while hens seen in treatment R, and at all treatments in locations ControlR and Away, were all similar (Wald_6_/ndf = 13.20 by GLMM; *p* < 0.001) (Figure 8b). Walking/running decreased marginally with bird age (predicted means ± SE logit (back-transformed proportions) 34 weeks −2.10 ± 0.09 (0.11), 52 weeks −2.38 ± 0.11 (0.08), 70 weeks −2.59 ± 0.13 (0.07); Wald_2_/ndf = 4.47 by GLMM; *p* = 0.011).

There were very few counts of birds seen dustbathing, feather pecking, aggressive pecking, perching, or in ‘other’ behaviours, so these are not reported further.

### 3.3. Feather Scores

Feather scores were low (i.e., little damage) at bird ages 34 and 52 weeks, with only tails having some damage (Table 6). Feather scores were highest at age 70 weeks, with a mean total feather score of 2, but mean feather scores at each body site were each less than 1. The prevailing effects on feather score were due to age and (when examining the individual scores) from the tails (where scores were highest; scores were lowest at breast, and in between for neck, back, and wings) (site.bird.age interaction, Wald_8_/ndf = 15.42 by LMM; *p* < 0.001). Many interactions would not converge due to sparse data or were not significant (*p* > 0.05) in the GLMMs applied to individual sites data, so this is not reported further.

Total mean feather scores were significantly affected by the interaction of treatment and age, whereby feather scores were lowest for B hens at 52 weeks of age, but were higher than PF at 70 weeks of age (*F*_6,43_ = 3.8 by LMM; *p* = 0.004) (Figure 9a), but in reality, these differences were small (back-transformed means: age 52 weeks, B 0.14 versus other treatments (range) 0.19–0.24; age 70 weeks, B 1.80 versus PF 1.33), and furthermore, the difference between 52 and 70 weeks may have been influenced by the lack of data from four out of eight flocks at age 70 weeks. There was a further interaction between age and time (Figure 9b), with no differences between times at ages 34 or 52 weeks, but with more hens seen with poorer feather scores at time ≥1 compared to time −1 at 70 weeks (*F*_4,2058_ = 4.7 by LMM; *p* = 0.001), but again, in reality, differences were small (back-transformed means age 70 weeks: 1.35–1.69) and may have been influenced by missing data from half of the flocks at age 70 weeks.

### 3.4. Replacement Frequency and Cost

Enrichments were replaced regularly by the farms based on their judgement of depletion. As a result, rates of replacement varied widely from flock to flock (Figure 10) apart from with PF, which was scattered twice a day in every flock (not shown). For example, replacement of PB pairs was highest in flocks A and B (which were on the same farm). Replacement of ropes was understandably higher in the treatment R, where there were twice as many ropes as in B, PB, or PF, but was lowest in flocks A, B, E, and F in all quarters. When covariates on days since last replacement and the cumulative amounts of enrichments replaced were tested last in the above-reported statistical models of behaviour data, as would be expected, the more recently items had been replaced, the more interest was shown by the birds. These covariates were often statistically significant with estimated coefficients in the expected direction, but no further details of this modelling are reported, as these covariates were observational, and the full range of their scales was only sparsely represented in the data.

The estimated and actual rates of enrichment replacements, and the total costs for use, are shown in Table 7. Flocks studied here were followed to 70+ weeks of age; however, flocks are likely to be housed for longer than this, depending on production. Therefore, the following cost estimates were based on the actual mean rate of replacement shown, in 16,000-hen flocks housed from 16 to 80 weeks (ignoring varying rates in mortality), thus needing enrichments for 64 weeks. Note that flocks are often expected to be given a variety of enrichments. Here, we estimated the costs based on providing each enrichment per 16,000 hens. However, where required (e.g., by accreditation schemes), flock managers would have to choose combinations of the enrichments shown to determine the total cost per flock. For example, RSPCA Assured require two items of permanent, destructible enrichment for every 1000 hens [22], so two items below would have to be added together (and pelleted feed might not be permitted, if not considered permanent, despite it being of greater interest than ropes).

With all enrichments used, the mean replacement rate varied widely from flock to flock: standard deviation values were 33–40% of the mean values. However, it was still clear that while Lucerne bales, pecking blocks, and pelleted feed generated the most interest in hens, ropes were by far the cheapest enrichment to provide. The most expensive was pecking blocks, followed by pelleted feed, then bales.

## 4. Discussion

The expected benefits of providing destructible enrichments are to encourage birds to direct pecking behaviours away from other hens, fulfil natural behaviour, and improve feather cover. In this study, we considered both interacting (i.e., pecking, scratching, pulling) with the enrichments plus foraging behaviour in the litter (which excluded the PF scatter area in that treatment). While foraging behaviour alone showed little differences between treatments, apart from a drop in foraging with PF as hens were drawn to the scattered feed area, all the nonrope enrichments achieved the desired goal of encouraging interaction at the enrichments, which would hopefully benefit feather cover. However, feather-cover responses were unclear and probably exacerbated by the loss of data, plus hens were able to move out of popholes in one quarter, and re-enter the shed at another quarter, thus potentially mixing some birds between the treatments. Previous research suggests that bird mixing was unlikely to have a large effect on our feather score data, since only small proportions of flocks are typically seen on range [23], particularly with large (≥16,000) hen flocks [24]. Feather cover did worsen with age, as expected, but feather cover was generally good (overall total feather scores on average of 2 or less), which is highly desirable. It may be that since evidence of feather pecking (via feather scores) was low in these flocks, there were only small differences gained from different enrichments, and a better comparator would be to have a treatment with no enrichments at all. However, that was not possible in these commercial flocks, which were required to provide enrichments by the accreditation schemes. Another theory is that enrichments may benefit hens with access to an outdoor area less than barn-system hens, which have the same indoor design as free-range hens, but no range access. For example, Heerkens [25] found plumage damage was worse in commercial barn flocks than in free-range flocks. However, given previous evidence of small proportions of free-range hens using the range, and evidence of feather pecking in free-range hens [3,10,25] particularly where range use was low [26], then appropriate enrichments are still likely to benefit birds in this system.

In this study, ropes were least useful for hens, based on the lack of hens observed in the vicinity of, and interacting with, ropes. A high proportion of hens were seen standing/sitting in both the ControlR and Away locations with all treatments, but this was significantly lower with B, PB, and PF compared to R in the Enrich locations, probably related to the commensurate increase in birds interacting with enrichments (other than R) in Enrich, which occupied 0.370–0.599 of the mean proportion of hens observed. This suggests that, of the four enrichments studied, ropes were no more attractive to hens than no enrichment at all. In contrast, interacting directly with the enrichments was significantly greater with PB and B at all observation times, while PF interaction peaked at feed scatter (with a concurrent decline in hens in the vicinity of, but not engaging with, the PF enrichment), but declined within an hour, probably because most pellets had been consumed by then, but hens were still showing an interest in PF at other times (−1, ≥1) compared to R or areas where there were no enrichments.

Previous work showed that using string (white polypropylene bailing twine) reduced both gentle and severe feather pecks, and elicited pecking at the string, in layer chicks housed on litter floors from 1–63 days of age, but also that the later that strings were introduced, the more negligible the effects [11]. It is unknown if hens used in our study had experienced string during rearing, but if their first introduction was upon entering the laying house, then it may be less surprising that they did not interact with ropes. String (or rope) may be more effective in wire-floor systems (e.g., enriched cages) where hens will encounter them more easily (due to the smaller overall space) than in litter-based systems where the much larger litter-covered floor, and other enrichments presented there, encourage foraging behaviour more effectively due to their size and/or position. For example, McAdie et al. [11] found that with hens reared in cages, presenting string at point of lay was effective at reducing feather damage. Of the four enrichments used here, all were destructible, but ropes had no nutritional value (unlike the other three). This, combined with their comparatively low volume (15.1 cm^3^ each versus hay bales of 102,375 cm^3^, and pairs of pecking blocks of 9868 cm^3^), may have combined to make them not only unattractive, but also comparatively difficult to locate. Given the lack of interest around ropes shown here, there was little supporting evidence to suggest that increasing the number of rope bundles would bring any benefit to hen behaviour, at least in free-range systems.

The mean counts of birds seen in any location, engaged in all behaviours, ranged from 6.8–17.2 birds. Given that the observation locations of 1 m diameter each provided an area of 7854 cm^2^, then on the basis of stocking density for hens in free-range systems of 9 hens/m^2^ (equivalent to 1111 cm^2^ per hen), this would have comfortably allowed space for 7 hens. In locations where there were ropes (all treatments in ControlR, and treatment R in Enrich) or no enrichments (Away) there were on average about the number of hens expected based on this stocking density, with 6.8–8.7 hens seen. In contrast, where there were B, PB, or PF enrichments (in location Enrich), we observed on average 10.8–17.2 hens, suggesting that birds were attracted to these enrichments. Bird attraction to the area was highest (and consistent) with PB, then B, whereas PF showed a decline in attraction outside of scattering, presumably because scattered feed was depleted. However, PF interest was still higher than that around ropes, suggesting that scattering of feed had long-lasting effects.

Adding feed or grain to the litter has been used previously to encourage foraging and reduce feather damage. Blokhuis and van der Haar [27] applied grain to litter pens of rearing pullets (40 g per pen of 12 pullets, three times per week), and found a significant increase in ground scratching compared to pullets supplemented with straw or nothing, plus the effects of grain carried over into the laying phase, in which hens had less feather damage compared to control hens. We saw a distinct rise in hens interacting with PF at the time of scatter, but also that this interacting behaviour was maintained long after the feed was presumably depleted, compared to the area around the ropes (ControlR), and also based on the high proportions of hens in the PF Enrich location. One criticism of using feed or grain is that it is not permanently available to hens, due to rapid depletion rates, but evidence here suggested that it elicited interacting behaviours more effectively than ropes (which were permanently available). However, in another study that examined relationships between management practices and feather pecking in over 111 free-range flocks in the UK, spreading feed on the floor was a significant risk factor for severe development of feather pecking [3]. In our study, effects of enrichments on feather cover were weak. Although hens interacted least with ropes, hens from rope treatment quarters showed intermediate levels of feather damage by age 70 weeks, similar to hens with pecking blocks (which was one of the most attractive enrichments to interact with), but damage overall was low across all treatments.

Providing enrichments comes at a cost to the producer, and must be balanced against benefits to the birds. If enrichments are impractical or costly, producers are unlikely to implement them [28]; however, if enrichments affect feather loss (which in turn increases feed costs, and can lead to mortality and reduced egg production) and/or fulfil an accrediting body’s requirements [22], they are more likely to be adopted. We could not conduct a full economic analysis in this study (e.g., examining the effects of enrichments on mortality, egg production, egg quality, and feed consumption), because mortality and egg production were not collected by shed quarter. However, based on enrichment costs alone, while rope was the cheapest enrichment by far over the lifetime of flocks, it was also the least effective in terms of effects on behaviour, and indistinguishable from behaviours observed in locations away from all enrichments in this study. In all shed quarters, we tested rope and another enrichment (or rope and rope, for control), but we did not test all combinations of the four treatments (e.g., B and PF, PB and PF, etc.) It may be that such combinations would have further benefits on behaviour and feather scores, but it is likely that the costs of these would be prohibitive to many producers. Therefore, given the requirements of some accreditation schemes for two different enrichments, rope + another is potentially a good compromise between interest for hens and reasonable costs, but it should be acknowledged that ropes are least likely to be of use.

## 5. Conclusions

Ropes are unsuitable enrichment for hens, in terms of encouraging interaction with the enrichment, but are inexpensive. In contrast, pecking blocks and alfalfa hay bales promoted interaction, but are comparatively expensive. Enrichments should be selected based on a balance between their efficacy and cost, in which case alfalfa bales are potentially the best choice from those studied here, but future studies that measure mortality, egg production, and egg quality according to enrichment type would be beneficial, to determine if enrichment costs are offset by other benefits.

## Figures and Tables

**Figure 1 animals-12-00995-f001:**
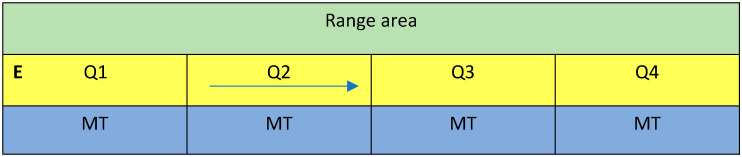
Overhead schematic of free-range hen sheds used in the study showing the four quarters, Q1–Q4. Q1 was always the quarter nearest to the annex, where staff would enter (**E**). The blue area is where the multitier (MT) structure was positioned, the yellow area was the litter, and birds could reach the range via popholes from the litter area. The arrow shows the direction that staff would walk, scattering litter from side-to-side in the relevant pelleted feed (PF) treatment/quarter.

**Figure 2 animals-12-00995-f002:**
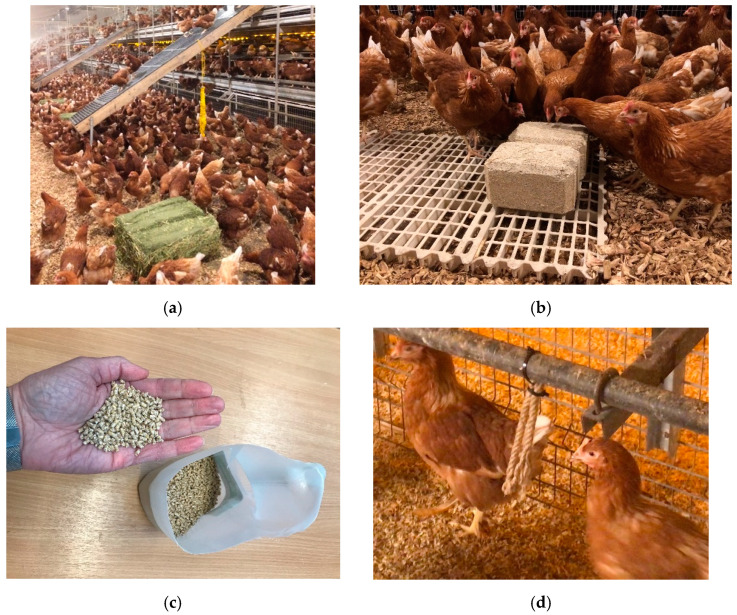
Enrichments: (**a**) alfalfa hay bales prior to hanging in the hay nets (yellow); (**b**) pecking blocks paired and on slats; (**c**) pelleted feed scattered from a plastic jug; (**d**) jute ropes.

**Figure 3 animals-12-00995-f003:**
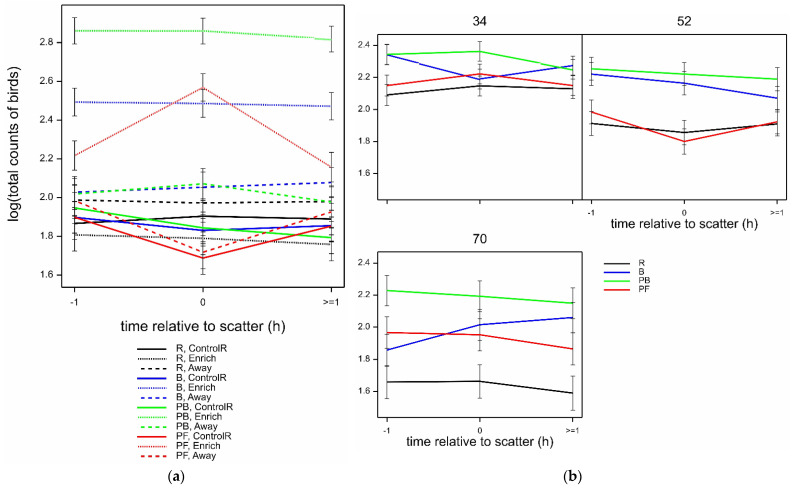
(**a**) Mean ± SE log (total counts of birds) observed over all behaviours in various locations (ControlR, Enrich, Away), according to enrichment treatments (R, B, PB, PF) and the time of observation relative to scatter of pelleted feed (−1, 0, ≥1), estimated from GLMM, with standard error (SE) bars shown. (**b**) Mean log (total counts of birds) observed over all behaviours with different enrichment treatments (R, B, PB, PF), according to bird age (34, 52, 70 weeks) and the time of observation relative to scatter of pelleted feed (−1, 0, ≥1), estimated from GLMM, with SE bars shown.

**Figure 4 animals-12-00995-f004:**
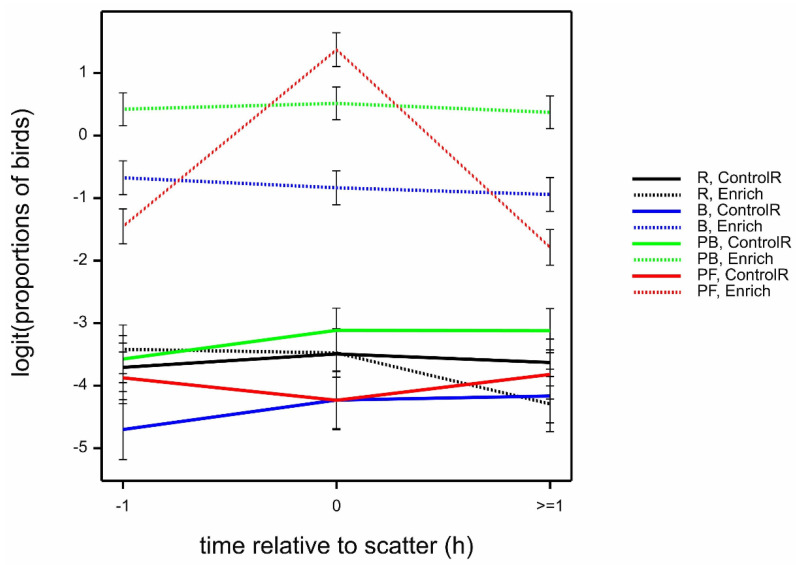
Mean ± SE logit (proportions of birds) observed interacting with enrichments, by time relative to scatter (−1, 0, ≥1) and location (ControlR, Enrich), estimated from GLMM. In all ControlR locations, the only enrichments present were ropes (R); in Enrich locations, there were ropes (R), bales + ropes (B), peck blocks + ropes (PB), or pelleted feed + ropes (PF).

**Figure 5 animals-12-00995-f005:**
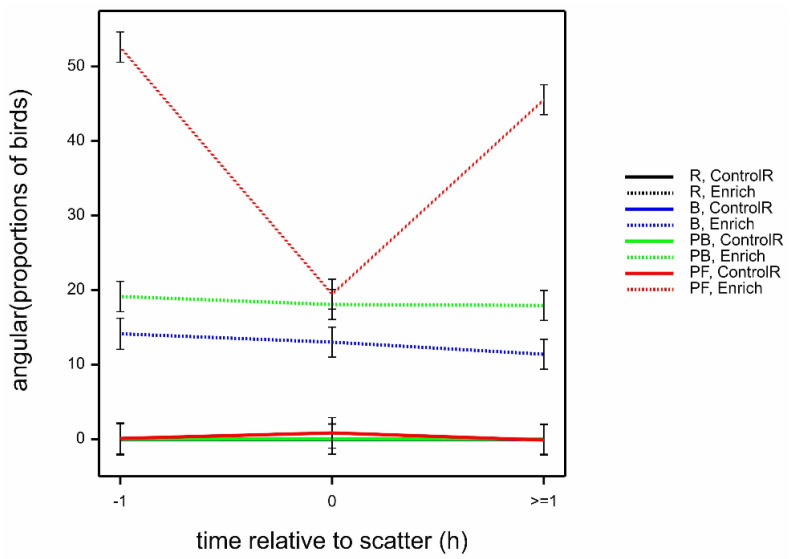
Mean ± SE angular (proportions of birds) observed at, but not interacting with, enrichments in Enrich and Control R locations, according to treatment (R, B, PB, PF) and time relative to scatter (−1, 0, ≥1), estimated from LMM. (Note that estimates are all 0 for R, Enrich and for R, B and PB at ControlR).

**Figure 6 animals-12-00995-f006:**
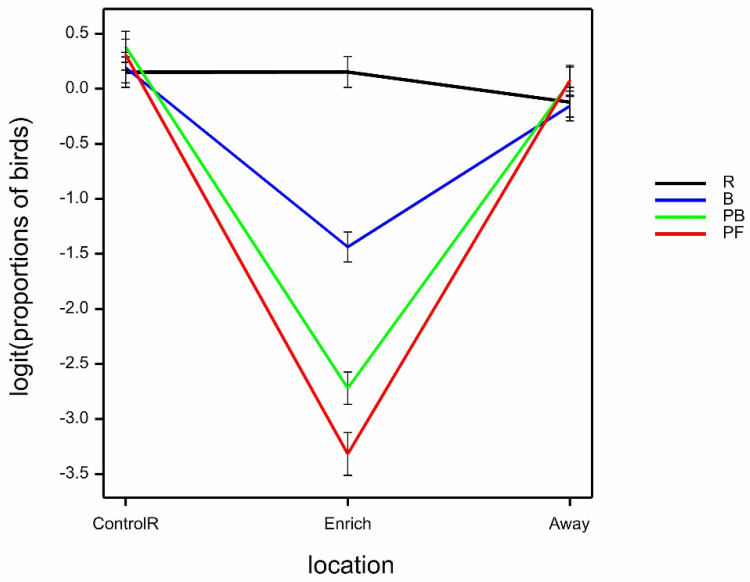
Mean ± SE logit (proportions of birds) observed in stand/sit behaviour, according to location (ControlR, Enrich, Away) and treatment (R, B, PB, PF), estimated from GLMM.

**Figure 7 animals-12-00995-f007:**
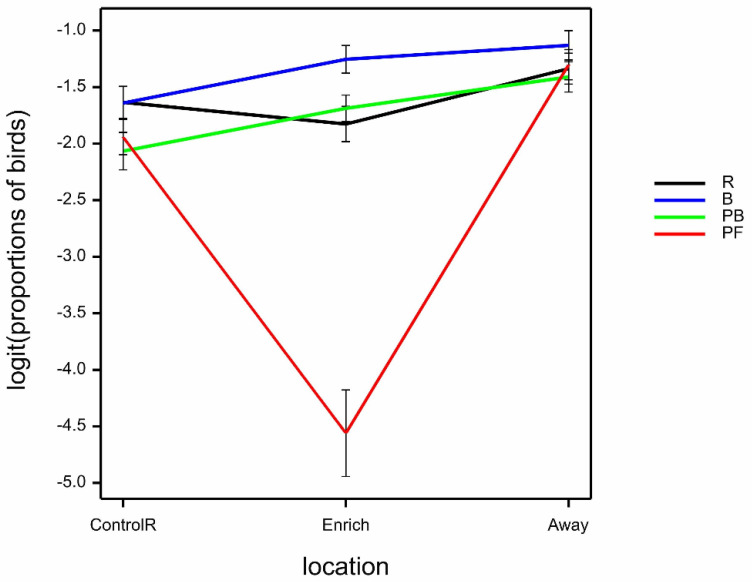
Mean ± SE logit (proportions of birds) observed in foraging behaviour, according to location (ControlR, Enrich, Away) and treatment (R, B, PB, PF), estimated from GLMM.

**Figure 8 animals-12-00995-f008:**
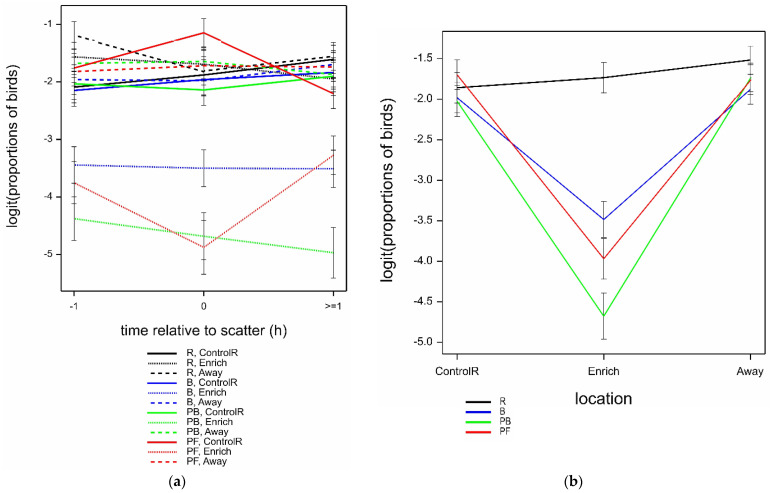
(**a**) Mean ± SE logit (proportions of birds) observed in walk/run behaviour, according to time (−1, 0, ≥1), location (ControlR, Enrich, Away) and treatment (R, B, PB, PF), estimated from GLMM. (**b**) Mean ± SE logit (proportions of birds) observed in walk/run behaviour, according location (ControlR, Enrich, Away) and treatment (R, B, PB, PF), estimated from GLMM.

**Figure 9 animals-12-00995-f009:**
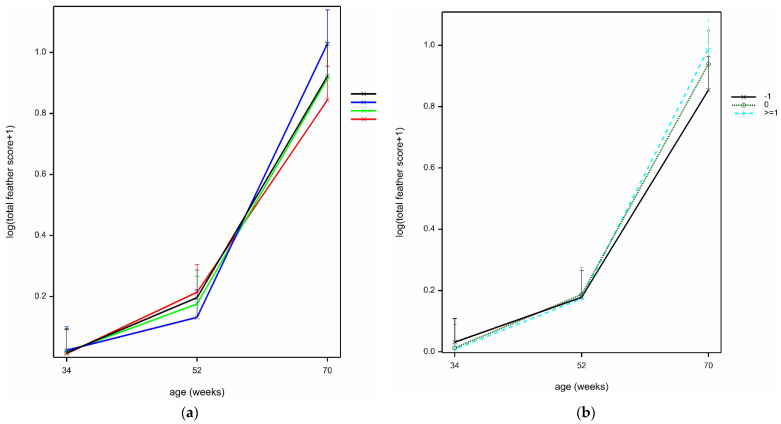
(**a**) Mean ± SE log (total feather score + 1) by age (34, 52, 70) and treatment (R, B, PB, PF), estimated from LMM. (**b**) Mean ± SE log (total feather score) by age (34, 52, 70) and time relative to scatter (−0, 0, ≥1), estimated from LMM.

**Figure 10 animals-12-00995-f010:**
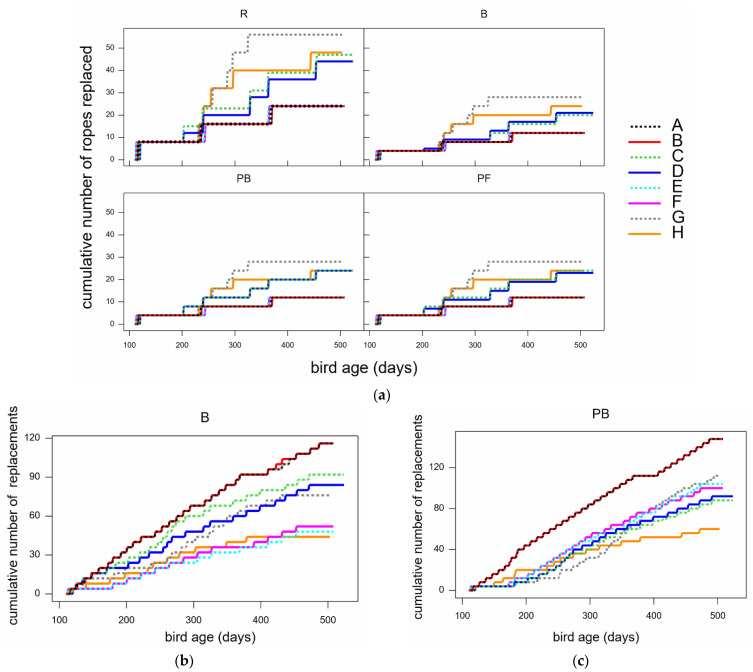
The cumulative number of enrichments replaced in each flock (A–H) over bird age (days). (**a**) Replacement of all ropes (in all treatments, R, B, PB, PF); (**b**) replacement of all hay bales (B); and (**c**) replacement of all pecking block pairs (PB).

**Table 1 animals-12-00995-t001:** Enrichment treatments (B, PB, PF, R), and their layout per flock, according to which quarter of the shed the items were provided in (quarters 1–4; quarter 1 was the section nearest the annex door).

Flock	Bales and Ropes (B)	Pecking Blocks and Ropes (PB)	Pelleted Feed and Ropes (PF)	Ropes Only (R)
A	Q1	Q2	Q3	Q4
B	Q3	Q1	Q4	Q2
C	Q2	Q4	Q1	Q3
D	Q4	Q3	Q2	Q1
E	Q3	Q4	Q1	Q2
F	Q2	Q1	Q4	Q3
G	Q1	Q3	Q2	Q4
H	Q4	Q2	Q3	Q1

**Table 2 animals-12-00995-t002:** List of mutually exclusive behaviours. The first two behaviours could not be assessed for location ‘away’ (because there were no enrichments there).

Behaviour
* Interacting with (e.g., peck, pull, scratch at) enrichment (or in litter where feed was scattered, PF treatment),
* At, but not interacting with, enrichment: birds were located within 1 m diameter of the enrichment, but were not in contact with it
Stand/sit: birds were holding still and performing no other behaviour
Forage: peck/scratch at litter (but not at location where feed is scattered, PF treatment)
Walk/run: birds were in locomotion
Dustbathe: birds were in a prone position, while raking litter with their beaks, or tossing/rubbing litter onto the plumage
Feather peck: gentle or vigorous pecks at the plumage of other birds, often repetitive until the target bird withdrew
Aggressive peck: forceful, downward pecks directed towards the head or neck
Perch: birds standing or sitting on perch rails
Other: any other behaviour

* Only collected at locations ControlR and Enrich.

**Table 3 animals-12-00995-t003:** The mean over scans of proportions of hens observed by location and treatment in various behaviours. All behaviours were mutually exclusive, and rows within location by treatment add up to 1.0. At ControlR, the only enrichment to interact with was rope; at Away, there were no enrichments. Figures in red are values ≥0.500; figures in blue are values between 0.100 and 0.499.

Location	Treatment	Behaviour									
		Interacting *	At But Not Interacting *	Stand/Sit	Forage	Walk/Run	Dustbathe	Feather Peck	Aggressive Peck	Perch	Other
ControlR	R	0.052	0.000	0.509	0.180	0.170	0.009	0.006	0.000	0.004	0.070
	B	0.033	0.000	0.525	0.203	0.141	0.010	0.004	0.000	0.015	0.069
	PB	0.060	0.000	0.552	0.153	0.147	0.006	0.005	0.000	0.006	0.071
	PF	0.032	0.005	0.564	0.143	0.184	0.004	0.006	0.001	0.006	0.055
Enrich	R	0.048	0.000	0.517	0.166	0.182	0.009	0.003	0.000	0.009	0.066
	B	0.370	0.094	0.218	0.235	0.047	0.002	0.004	0.000	0.000	0.030
	PB	0.599	0.111	0.083	0.169	0.020	0.000	0.001	0.001	0.000	0.016
	PF	0.378	0.437	0.063	0.027	0.038	0.001	0.002	0.001	0.000	0.053
Away	R	NA	NA	0.452	0.238	0.197	0.017	0.012	0.000	0.000	0.085
	B	NA	NA	0.445	0.266	0.170	0.023	0.011	0.001	0.003	0.082
	PB	NA	NA	0.489	0.217	0.191	0.008	0.005	0.002	0.000	0.088
	PF	NA	NA	0.512	0.218	0.190	0.019	0.001	0.002	0.000	0.058

* With enrichment; NA = not applicable, because there were no enrichments to interact with.

**Table 4 animals-12-00995-t004:** The mean ± SD over scans of total counts of birds observed over all behaviours, according to location (1 m diameter around the control ropes (ControlR), the enrichment (Enrich), or away from either (Away)) and enrichment treatment (i.e., ropes (R), bales (B), peck blocks (PB), or pelleted feed (PF)) provided in shed quarters. The estimated means (back-transformed from GLMM) are shown in brackets (which are adjusted for missing data).

Location			
Treatment	ControlR	Enrich	Away
R	7.4 ± 3.6 (6.6)	6.8 ± 3.1 (6.0)	8.2 ± 3.1 (7.2)
B	7.1 ± 3.4 (6.4)	13.1 ± 4.4 (12.0)	8.7 ± 3.3 (7.8)
PB	7.2 ± 3.6 (6.4)	17.2 ± 4.0 (17.2)	8.3 ± 3.1 (7.6)
PF	6.8 ± 3.4 (6.1)	10.8 ± 3.4 (10.1)	7.3 ± 3.0 (6.5)

**Table 5 animals-12-00995-t005:** Mean ± SE log (total counts of birds) (back-transformed shown in parentheses) observed over all behaviours by age (34, 52, and 70 weeks) and location (ControlR, Enrich, Away), estimated from GLMM.

	34 Weeks	52 Weeks	70 Weeks
ControlR	1.94 ± 0.10 (7.0)	1.90 ± 0.11 (6.7)	1.73 ± 0.12 (5.6)
Enrich	2.47 ± 0.10 (11.9)	2.33 ± 0.10 (10.2)	2.27 ± 0.11 (9.7)
Away	2.25 ± 0.10 (9.5)	1.91 ± 0.11 (6.7)	1.79 ± 0.12 (6.0)

**Table 6 animals-12-00995-t006:** Mean ± SD feather score by bird age and body location (overall treatments and flocks) and mean ± SD total feather score (FS).

	34 Weeks	52 Weeks	70 Weeks
Neck	0.000 ± 0.000	0.003 ± 0.053	0.398 ± 0.536
Back	0.000 ± 0.000	0.001 ± 0.037	0.362 ± 0.520
Tail	0.024 ± 0.153	0.257 ± 0.444	0.664 ± 0.495
Breast	0.000 ± 0.000	0.000 ± 0.000	0.145 ± 0.358
Wings	0.000 ± 0.000	0.000 ± 0.000	0.436 ± 0.580
Total FS	0.024 ± 0.153	0.261 ± 0.452	2.004 ± 1.790

**Table 7 animals-12-00995-t007:** The estimated and actual mean rate of replacement for the four enrichments in eight flocks, with standard deviation (SD) given, and the total cost of using each enrichment in a flock of 16,000 hens, housed for 64 weeks (16–80 weeks of age), based on the actual mean rate of replacement seen here. Costs do not include local taxes or shipping.

	Bales	Pecking Blocks	Pelleted Feed	Rope
Estimated replacement	21 days	10 days	Twice a day	180 days
Mean replacement (*n* = 8)	21.9 days	14.9 days	Twice a day	96.6 days
SD (*n* = 8)	8.8	4.9	0	36.3
Cost of 1 item (GBP)	GBP 6.50/bale	GBP 7.00/block	GBP 8.38/20 kg bag (GBP 419/tonne)	8.295 *p*/30 cm (GBP 27.65/100 m reel)
No. required for 16,000 hens	16	32	16 kg	16
Cost as 1 enrichment for 16,000 hens	GBP 104.00	GBP 224	GBP 6.70	GBP 1.33
Number of times item would need replacing in 64 weeks	20	30	448	5
Total cost	GBP 2080	GBP 6720	GBP 3008 *	GBP 6.64

* Only 7168 kg needed, but feed can only be bought in bags of 20 kg, so 7080 kg = GBP 3008.

## Data Availability

The data presented in this study are openly available in Zenodo at https://doi.org/10.5281/zenodo.5996957 (accessed on 31 March 2022).

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
