# Peer review of "Do Hens Use Enrichments Provided in Free-Range Systems?"

_animals, 2022, doi:10.3390/ani12080995_

Round 1
Reviewer 1 Report
The authors should be commended for conducting research using on-farm animals in real environments. There are so many different factors across and between farms that could not be controlled for so it is clear this was a challenging research trial.
Abstract:
Line 23: Hens is the subject of the sentence so you cannot write "their efficacy requires investigation". Could write "the efficacy of enrichments requires investigation".
Introduction:
Page 2, Line 72 - what do the authors hypothesize?
Page 2, Lines 72-76: "The work demonstrated... " This is summary and conclusions and should not appear in the introduction.
What have other studies reported about the impacts of enrichments with nutritional benefit on egg production, egg quality, feed consumption or other measures of productivity that could be affected by addition of nutritional enrichments?
Materials and methods:
Was the pelleted feed enrichment the same feed formulation as daily feed? How have you accounted for the impact of nutritive enrichment on egg production and quality (both PB and PF have calcium content, protein content)?
It is unclear why rope was chosen given that these are non-cage environments and rope does not fit the fundamental characteristics that elicit foraging behaviour (fall apart on pecking and have nutritional benefit)?
Page 4, Lines 149-151: Clarify the error noted
Page 5, Line 155: Why were behavioural observations not conducted at an earlier age (shortly after pullets arrived) so that a baseline of feather cover could be established at the time enrichments were installed - to establish whether there were Q differences in feather cover from the begin of the study?
Page 5, Table 2: Define dustbath, feather peck, aggressive peck
Page 5, Lines 178-184: why were birds sampled for feather score from the Away location? Birds who are not foraging (and hence are in the Away location) may be subordinate birds who are prevented from foraging by more aggressive birds who feather peck the subordinate birds. Sampling is therefore not random.
Additionally, only 10 birds per Q (or 10 birds per 4000) were sampled for feather cover. This does not seem a sufficient sample size for a powerful statistical analysis. Was a power calculation conducted to determine sample size required?
Page 5, Line 187: should read "...and no behaviour data could be collected..."
Page 6, Lines 204-211: should be in section 2.3 and not first reported in statistical analysis
Page 6, Lines 212-213 are part of results, not methods and lines 214-215 should be part of feather score section in methods, not statistical analysis.
Line 224: closing bracket needed
Page 7, Lines 256-259: Actual flock visits... should be part of methods and not results.
There are a lot of results displayed in graphs and a lot of data is shown without really addressing the key findings.
Why have the authors not conducted a correlation analysis between foraging behaviour and feather score?
Conclusions:
The authors cannot conclude from this study that PB and alfalfa hay bales are comparatively expensive as a full economic analysis has not been conducted in this study. While the cost to purchase the enrichments is shown, the benefits have not been calculated and economically assessed. Have the enrichments affected egg production, egg quality, feed consumption, morbidity and mortality. These are all economic considerations that might alter the costs and benefits.
Author Response
The authors should be commended for conducting research using on-farm animals in real environments. There are so many different factors across and between farms that could not be controlled for so it is clear this was a challenging research trial.
- Thank you. On one hand we could control things better at our own research facility, but this does not work on the scale of commercial farms so it is certainly a trade-off...
- Thank you very much for taking the time to read our paper and contribute useful comments.
Abstract:
Line 23: Hens is the subject of the sentence so you cannot write "their efficacy requires investigation". Could write "the efficacy of enrichments requires investigation".
- Done (line 24)
Introduction:
Page 2, Line 72 - what do the authors hypothesize?
- Added to line 87-89
Page 2, Lines 72-76: "The work demonstrated... " This is summary and conclusions and should not appear in the introduction.
- We agree that it is odd, however the ANIMALS template states (in the Introduction section) “Finally, briefly mention the main aim of the work and highlight the principal conclusions.” However, we are happy to take it out! Now removed from line 90-94. It is up to the journal editor if they would prefer this is left in.
What have other studies reported about the impacts of enrichments with nutritional benefit on egg production, egg quality, feed consumption or other measures of productivity that could be affected by addition of nutritional enrichments?
- Schreiter et al (2020) found no effect of enrichments (alfalfa hay and pecking stones) on daily feed consumption, body mass, mortality, or egg shell breaking strength, but did find a that enrichments affected albumen quality. Cronin et al (2018) found no effect of straw enrichment on laying performance. Blockhuis & Van der Haar 1992 found that supplying grain in rear reduced pecking at food in rear (but not necessarily lowered feed intake) and this persisted in lay, when the enrichment was no longer given. Steenfeldt et al 2007 found egg production increased with two out of three foraging materials, but intake of these made up 33-48% of total feed intake, Other papers using destructible/edible enrichments (e.g. Aerni et al 2000, Zepp et al 2018) did not study feed intake or egg characteristics. We have highlighted these points in the Introduction, lines 75-84
Materials and methods:
Was the pelleted feed enrichment the same feed formulation as daily feed? How have you accounted for the impact of nutritive enrichment on egg production and quality (both PB and PF have calcium content, protein content)?
- The pelleted feed was not the same formulation as daily feed – they were different manufacturers (each farm got its feed from different sources). We did not account for the impact of nutritive enrichment on eggs, since a) the additional nutrients from using enrichments is very small (based on predicted and actual use per hen, e.g. 1 g per hen per day or less). We did not consider examining egg production or quality due to predicted low intake. We have added information to line 170-172 to highlight this.
It is unclear why rope was chosen given that these are non-cage environments and rope does not fit the fundamental characteristics that elicit foraging behaviour (fall apart on pecking and have nutritional benefit)?
- The work was undertaken to reflect commercial practice, and this is a common enrichment used by producers. It has also been shown in previous studies to reduce plumage damage (Eg McAdie et al 2005). It is in fact somewhat destructible (its shape alters as it frays; cf cones and CDs) however the use of nutritive enrichments has risen in recent years, thus the comparison. We have added something to lines 70-73 to this effect.
Page 4, Lines 149-151: Clarify the error noted
- Now line 174, corrected
Page 5, Line 155: Why were behavioural observations not conducted at an earlier age (shortly after pullets arrived) so that a baseline of feather cover could be established at the time enrichments were installed - to establish whether there were Q differences in feather cover from the begin of the study?
- We presumed that all pullets in all quarters would have had similar feather cover prior to the influence of any enrichments, since per flock they are reared in one large group, and then split into quarters on arrival at the laying farm. Feather cover changes are gradual and cumulative, with feather pecking starting to be obvious from around peak lay (thus the 34 week sample date). In our study, feather scores were generally good (i.e. low) [although other factors (loss of data, mixing) were at play] particularly at 34 weeks (where all data could be collected at all farms), so we have no reason to suspect that there might have been differences on arrival. More specifically only 2.3% of birds had feather score 1 for tail at 34 weeks, with feather scores 0 for all other sites, so earlier scores are likely all 0 and so would not provide any variation in baselines.
Page 5, Table 2: Define dustbath, feather peck, aggressive peck
- Now added to Table 2
Page 5, Lines 178-184: why were birds sampled for feather score from the Away location? Birds who are not foraging (and hence are in the Away location) may be subordinate birds who are prevented from foraging by more aggressive birds who feather peck the subordinate birds. Sampling is therefore not random.
- We chose to feather sample birds from one of the three locations being observed already, and chose ‘Away’ since birds are more likely to be spread out here and not clustered together around enrichments (which would make viewing some body areas more difficult). If your theory about feather pecked birds being more likely to be in this area, then this would be a good reason indeed to sample from here, as a worst-case scenario with regards to feather cover. Since enrichments should fulfil foraging, it should reduce feather pecker behaviour, which would subsequently reduce signs of feather pecking in victims.
Additionally, only 10 birds per Q (or 10 birds per 4000) were sampled for feather cover. This does not seem a sufficient sample size for a powerful statistical analysis. Was a power calculation conducted to determine sample size required?
- 10 birds were sampled per quarter at each sampling time (-1, 0, >1) so in fact 30 hens per quarter per age were sampled at each farm, which is ~240 birds per enrichment at each age. We have moved the order of text around in line 210 so this is clearer. Power calculations were not carried out. Because the data is ordinal with few levels, or analysed as discrete counts, and because the appropriate analysis is a mixed model with a complex hierarchy of random effects, a bespoke simulation program would have to be written to do this properly. Approximate power calculations based on a simpler approach would not be sufficiently accurate for this data. In any case, some fairly negligible and inconsistent interactions in the complex model were statistically significant, which would indicate that if there were biologically relevant effects of the different enrichments in the collected feather score data, they would likely have been detected from a study this size.
Page 5, Line 187: should read "...and no behaviour data could be collected..."
- Amended line 214
Page 6, Lines 204-211: should be in section 2.3 and not first reported in statistical analysis
- We disagree with this, and the subsequent comment, as what is reported here is not to do with the way data was collected, but how it was subsequently processed in different ways for statistical analyses – this is a description of the various response measures used, needed here in order to understand the appropriateness of the statistical models.
Page 6, Lines 212-213 are part of results, not methods and lines 214-215 should be part of feather score section in methods, not statistical analysis.
- We disagree, as it was only after doing the analyses that we could see that results were similar, and chose to omit them to avoid repetition. The statistical analysis of both where fully carried out though and so are mentioned here because the interpretation of two measures differs and so it would have been of interest if they had shown differing results.
Line 224: closing bracket needed
- Not sure where: the part that starts with “i.e.” has closing bracket after “-p”:
“(i.e., for proportion p, loge (p/1-p))”
Page 7, Lines 256-259: Actual flock visits... should be part of methods and not results.
- Now moved to line 175-178
There are a lot of results displayed in graphs and a lot of data is shown without really addressing the key findings. Why have the authors not conducted a correlation analysis between foraging behaviour and feather score?
- The lack of variation in feather scores means that any association between feather scores and behaviour is likely to be marginal. Furthermore, a simple correlation measure between them based on the raw data is not appropriate, as feather scores are discrete, and, also P values for correlations would be invalid as they do not account for the structure in the data (random and fixed effects). However, we did summarise all the data by calculating means at the level of each flock by quarter by age, and examine on the summarised data correlations and scatter plot matrices over all ages and at each age. The correlations between the mean of the total feather scores and mean proportions of behaviours are shown below. Whilst feather scores are positively correlated with stand/sit and negatively correlated with walk/run and forage, the scatterplot matrices (coloured by age or enrichment) show that the relationships are not strong and not associated with enrichments. Because of this, and because to fully investigate the relationship correctly further statistical modelling would be needed, we have not included correlations in the paper.
(images are shown in the attached file)
Conclusions:
The authors cannot conclude from this study that PB and alfalfa hay bales are comparatively expensive as a full economic analysis has not been conducted in this study. While the cost to purchase the enrichments is shown, the benefits have not been calculated and economically assessed. Have the enrichments affected egg production, egg quality, feed consumption, morbidity and mortality. These are all economic considerations that might alter the costs and benefits.
- We have reworded the aim (line 89) to say ‘estimated’ rather than ‘investigated’, and reworded lines 590-594 to show that this is based on limited information only,

Reviewer 2 Report
This study is generally clearly written and is of interest to producers and researchers seeking to understand what enrichments on-farm are actually effective.
I do query the feather scores when hens could move between treatment quarters. Can you really compare among treatments if hens were not fixed in their treatment locations?
I also think there could be more text/discussion around the range aspect of a free-range system and the benefit of enrichments inside for hens that have access to an outdoor area. Would you see much greater use/benefit of enrichments in enclosed litter based systems?
More specific comments are listed below.
Simple summary and abstract: between the two of these, I gather the shed was divided internally (by the researchers or there were actually physical divides?), but that hens could mix? Was this mixing after they went out onto the range and then were free to come back into any quarter? This seems quite important for feather scores and would be good to have this clear. Also, could there be some indication of bird ages during the study?
Line 23: (or somewhere else in the abstract). It would be helpful to make it more explicit that enrichments were provided inside the shed rather than out on the range. Other studies with enrichments in free-range systems have placed them in the range area rather than the shed.
Lines 27, 31, 33. It would be easier for the reader if the enrichments were listed in the same order when defined, then abbreviated.
Line 47: function or functioning?
Line 52: what about the recent Rudkin paper. Feather pecking and foraging uncorrelated – the redirection hypothesis revisited. Is it worth including this and the alternative hypotheses for feather pecking and foraging?
Line 60: could you mention pullet rearing here as feather pecking may start before the birds even arrive at the layer sheds.
Line 69-76: This summary of results at the end of the introduction seems odd. I’ve not seen this done before, the summary would typically be placed at the beginning of the discussion.
General introduction comment: The intro is very short and missing some of the literature on enrichment in free-range hens. E.g., ‘Effect of pecking stones and age on feather cover, hen mortality, and performance in free-range laying hens’. It may also be helpful for the reader to have some text clarifying that enrichment can be provided inside or out on the range in free-range systems. This would be quite distinct from barn systems where there is only the inside shed. Increased ranging is associated with reduced feather damage, so enrichments sometimes aim to address welfare issues via that mode (enrich the range, more hens outside pecking, reduce feather pecking), rather than enriching inside the shed. What was the rationale for enriching inside rather than outside in the current study? Line 58 you state there is probably not a lot of motivation to forage in the litter inside, I would agree, but what about all the foraging opportunities outside? I feel there is a whole aspect missing without addressing the clear difference between barn and free-range (that you have lumped together in line 52 as alternative systems) and why enrichments inside may still be beneficial for systems with outdoor access.
Line 79: double-checking the May 2019 date is correct given the AE number is listed as 2018?
Lines 81-90: Birds were restricted inside the shed, but could they access any part of the range outside or was that also divided into quarters? Could there be a description of what the range area looked like? I would think the quality of the range and range use by the hens would be a key factor to understanding the relationship between feather pecking and indoor enrichments.
Line 102: what age did pullets arrive?
Line 148: space needed after (c)
Line 150: remove error text
Line 155: ‘due to take place’ seems like strange wording, did they not actually take place at those ages? (I see now is this related to line 185?) Maybe this could be made clearer earlier?
Line 155 onwards: what age did the pop-holes open for range access? Were the pop-holes open or closed during the behaviour observations?
Line 270: ‘The mean over’ was confusing at first, would ‘mean across’ be easier for the reader?
Line 347: check font
Figures 6, 7, and 8: what values are these in relation to hen age? Is this across all ages?
Lines 406 and 413: not sure if the words ‘in reality’ are needed to make the point that statistical differences were likely not biologically relevant
Line 469-471: this detail should be made clearer in the methods, and is a key factor in interpretation of the feather scores. I’m not convinced the feather scores are valid for comparing between treatments if the hens could move between treatments. You state ‘potential mixing’ but what evidence is there that hens would return to the same quarter once they went outside? What would be more likely, hens would return through the same pop-holes, or that hens would be mixing? If hens mixed, then I don’t think you can draw conclusions on feather scores per treatment.
Lines 495-501: could this sentence be reworded/split? It is difficult to follow. I think it should be ‘where’ not ‘were’ on line 498
Line 509: you state ‘at least in free-range systems’ and I agree this is a key point. Could you expand upon the benefit and use of indoor foraging enrichments for hens that have access to an outdoor area, and the attractiveness of the outdoor area? Does the accreditation require enrichments to be presented inside? Or is there a choice whether the enrichments are inside the shed or out on the range?
Author Response
This study is generally clearly written and is of interest to producers and researchers seeking to understand what enrichments on-farm are actually effective.
Thanks very much for taking the time to review our paper, it is much appreciated.
I do query the feather scores when hens could move between treatment quarters. Can you really compare among treatments if hens were not fixed in their treatment locations?
- Previous research has shown that that in large flocks, only a small percentage of hens are seen out on the range, so we have added some text to that effect to clarify this is unlikely to have had a large effect on our data, lines 505-508
I also think there could be more text/discussion around the range aspect of a free-range system and the benefit of enrichments inside for hens that have access to an outdoor area. Would you see much greater use/benefit of enrichments in enclosed litter based systems?
- That’s a valid point but given the small proportion that go on range (for various reasons – to be near food, due to adverse nature of open range etc), and the considerable proportions of free-range hen flocks that show feather pecking behaviour, then enrichment inside is still important. We have added something to lines 514-521
More specific comments are listed below.
Simple summary and abstract: between the two of these, I gather the shed was divided internally (by the researchers or there were actually physical divides?), but that hens could mix? Was this mixing after they went out onto the range and then were free to come back into any quarter? This seems quite important for feather scores and would be good to have this clear. Also, could there be some indication of bird ages during the study?
- Now added clarification to lines 104-108
- Bird ages are stipulated in section 2.3 (no change)
Line 23: (or somewhere else in the abstract). It would be helpful to make it more explicit that enrichments were provided inside the shed rather than out on the range. Other studies with enrichments in free-range systems have placed them in the range area rather than the shed.
- Currently the abstract says (line 26) “Sheds were split into quarters, in which two enrichments (jute ropes (R) + other) were installed” so we felt it was clear here, but we have added to line 25-26 and to the Methods section at line 121-122 (section 2.2)
Lines 27, 31, 33. It would be easier for the reader if the enrichments were listed in the same order when defined, then abbreviated.
- We did aim to do this: we define Ropes first (line 26), which then appears line 29 in ControlR first, and then in order they are given B, PB, PF, which is the order that they are defined in line 27
Line 47: function or functioning?
- now amended to functioning, line 48
Line 52: what about the recent Rudkin paper. Feather pecking and foraging uncorrelated – the redirection hypothesis revisited. Is it worth including this and the alternative hypotheses for feather pecking and foraging?
- Mentioned in line 76 along with another reference
Line 60: could you mention pullet rearing here as feather pecking may start before the birds even arrive at the layer sheds.
- Now added, line 62
Line 69-76: This summary of results at the end of the introduction seems odd. I’ve not seen this done before, the summary would typically be placed at the beginning of the discussion.
- We agree that it is odd, however the ANIMALS template states (in the Introduction section) “Finally, briefly mention the main aim of the work and highlight the principal conclusions.” However, we have taken it out (line 90-94) We leave it to the editors to decide if it should remain in..
General introduction comment: The intro is very short and missing some of the literature on enrichment in free-range hens. E.g., ‘Effect of pecking stones and age on feather cover, hen mortality, and performance in free-range laying hens’. It may also be helpful for the reader to have some text clarifying that enrichment can be provided inside or out on the range in free-range systems. This would be quite distinct from barn systems where there is only the inside shed. Increased ranging is associated with reduced feather damage, so enrichments sometimes aim to address welfare issues via that mode (enrich the range, more hens outside pecking, reduce feather pecking), rather than enriching inside the shed. What was the rationale for enriching inside rather than outside in the current study? Line 58 you state there is probably not a lot of motivation to forage in the litter inside, I would agree, but what about all the foraging opportunities outside? I feel there is a whole aspect missing without addressing the clear difference between barn and free-range (that you have lumped together in line 52 as alternative systems) and why enrichments inside may still be beneficial for systems with outdoor access.
- Elaborated on the literature cited and some more literature added, including the Iqbal paper referred to above lines 76-84
- The types of enrichments we were studying are provided inside of commercial sheds, where most of the birds are, and to protect the enrichments from contamination from wild birds, rain (which would lead to mould) etc. The accreditation scheme that the farms we worked with require that these enrichments are provided inside the shed.
- Range areas can be ‘enriched’ by ensuring there is plenty of vegetation, shelters, trees and bushes, and outdoor-appropriate drinkers (that cannot be contaminated by wild birds) but that was not the focus of our study,
- Text added to line 62-63 to include pullets, and clarify the focus is inside the shed
Line 79: double-checking the May 2019 date is correct given the AE number is listed as 2018?
- It is correct. We submitted the ethics form in late 2018 (thus the AE number) but the study didn’t start until spring 2019
Lines 81-90: Birds were restricted inside the shed, but could they access any part of the range outside or was that also divided into quarters? Could there be a description of what the range area looked like? I would think the quality of the range and range use by the hens would be a key factor to understanding the relationship between feather pecking and indoor enrichments.
- It was not possible to split the range into quarters –that would be a major investment in fencing (the range areas go for some distance) that we could neither afford nor were farm owners willing to do this.
- We did not investigate the range systematically, so cannot add information in good faith.
Line 102: what age did pullets arrive?
- Added line 101-102
Line 148: space needed after (c)
- Done (Figure 2 heading)
Line 150: remove error text
- Done
Line 155: ‘due to take place’ seems like strange wording, did they not actually take place at those ages? (I see now is this related to line 185?) Maybe this could be made clearer earlier?
- Now moved the actual flock visits text to this section, see lines 179-182
Line 155 onwards: what age did the pop-holes open for range access? Were the pop-holes open or closed during the behaviour observations?
- The accreditation scheme insists that they are opened within 3 weeks of arrival, thus by age 18-19 weeks. Popholes were open during observations – this has been added (lines 183-184).
Line 270: ‘The mean over’ was confusing at first, would ‘mean across’ be easier for the reader?
There are many instances throughout the paper where we use over, and across could be used instead. We discussed this among 2 independent colleagues (not co-authors) and they agreed it is sufficiently clear as it is.
Line 347: check font
- Well spotted, thanks, now altered.
Figures 6, 7, and 8: what values are these in relation to hen age? Is this across all ages?
- These are across all ages. Like the other graphs, where the factor is not specified, it is across all, e.g. sampling time (-1, 0 >1), locations, ages etc. Most of these tables and graphs for effects throughout most of the results are estimated from statistical models and all captions make that clear. It is usual to give estimates of main effects and interactions, when they are significant, averaging appropriately over other fixed and random effects in the models – that is just what the models do, and, most importantly, where there is missing data from a balanced design, a mixed model will average sensibly removing any biases caused by the missing data (whereas raw data summaries would not). It is not usual to state that this averaging is happening and what is being averaged over.
- Lines 406 and 413: not sure if the words ‘in reality’ are needed to make the point that statistical differences were likely not biologically relevant
That is exactly why we have added them, as people often latch onto the fact that something is (or isn’t) significant without considering the relevance.
Line 469-471: this detail should be made clearer in the methods, and is a key factor in interpretation of the feather scores. I’m not convinced the feather scores are valid for comparing between treatments if the hens could move between treatments. You state ‘potential mixing’ but what evidence is there that hens would return to the same quarter once they went outside? What would be more likely, hens would return through the same pop-holes, or that hens would be mixing? If hens mixed, then I don’t think you can draw conclusions on feather scores per treatment.
- Now added text to lines 505-508 to explain this is likely to have very small effects.
Lines 495-501: could this sentence be reworded/split? It is difficult to follow. I think it should be ‘where’ not ‘were’ on line 498
- Now shortened to be clearer line 541-543
Line 509: you state ‘at least in free-range systems’ and I agree this is a key point. Could you expand upon the benefit and use of indoor foraging enrichments for hens that have access to an outdoor area, and the attractiveness of the outdoor area? Does the accreditation require enrichments to be presented inside? Or is there a choice whether the enrichments are inside the shed or out on the range?
- Answered above – this is standard practice in the UK, required by the accreditation scheme, and probably due to most hens staying indoors.
Round 2
Reviewer 1 Report
-Check spelling of Schreiter et al. line 79
-Would still like to see what the authors think should follow from this study - eg. a comment in the conclusion of the paper indicating that future studies should examine whether there are economic benefits (improved egg quality or production, reduced morbidity or mortality) to providing enrichments that promote interaction (such as pecking blocks or alfalfa hay bales or pelleted feed) to determine whether the improvements to production would help to offset the cost of providing such enrichments (and might therefore encourage producers to provide interactive enrichments)
Author Response
Thank you for taking a further look at our revised paper.
line 79 typo corrected
Section added to conclusion "...but future studies that measure mortality, egg production, and egg quality according to enrichment type would be beneficial, to determine if enrichment costs are offset by other benefits. "